# Mining NCBI Sequence Read Archive Database: An Untapped Source of Organelle Genomes for Taxonomic and Comparative Genomics Research

**Vahap Eldem *** and **Mehmet Ali Balcı**

Department of Biology, Faculty of Science, Istanbul University, Istanbul 34134, Turkey; balci.mehmetali@ogr.iu.edu.tr
* Correspondence: vahap.eldem@istanbul.edu.tr; Tel.: +90-212-455-5700 (ext. 15086)

**Abstract:** The NCBI SRA database is constantly expanding due to the large amount of genomic and transcriptomic data from various organisms generated by next-generation sequencing, and re-searchers worldwide regularly deposit new data into the database. This high-coverage genomic and transcriptomic information can be re-evaluated regardless of the original research subject. The database-deposited NGS data can offer valuable insights into the genomes of organelles, particularly for non-model organisms. Here, we developed an automated bioinformatics workflow called "OrgaMiner", designed to unveil high-quality mitochondrial and chloroplast genomes by data mining the NCBI SRA database. OrgaMiner, a Python-based pipeline, automatically orchestrates various tools to extract, assemble, and annotate organelle genomes for non-model organisms without available organelle genome sequences but with data in the NCBI SRA. To test the usability and feasibility of the pipeline, "*mollusca*" was selected as a keyword, and 76 new mitochondrial genomes were *de novo* assembled and annotated automatically without writing one single code. The applicability of the pipeline can be expanded to identify organelles in diverse invertebrate, vertebrate, and plant species by simply specifying the taxonomic name. OrgaMiner provides an easy-to-use, end-to-end solution for biologists mainly working with taxonomy and population genetics.

**Keywords:** next-generation sequencing; sequence read archive; organelle genome; systematics





## 1. Introduction

Whole-genome-sequencing (WGS) and RNA-sequencing (RNA-Seq) studies that employ new or third-generation sequencing approaches generate millions of reads and vast amounts of genomic data per sample. The high- or low-coverage WGS and RNA-Seq approaches have become routine procedures for conducting a variety of studies, including population genetics studies [1], comparative genome analyses [2,3], and genome-wide association studies (GWASs) [4], as well as studies on clarifying the molecular mechanism of organ development [5,6], sex determination [7], understanding the physical effects of exposure to biological and chemical agents [8–10], and other gene expression-based research on non-model organisms [11]. Due to the vast amount of genomic and transcriptomic data generated, this high-coverage nucleotide information can be re-evaluated regardless of the original research subject. For instance, these high-volume genomic data can yield significant insights into the genomes of organelles that are abundantly present in tissues, particularly mitochondria and chloroplasts. In vertebrates, organs such as the skeletal muscle, heart, liver, kidney, brain neurons, theca cells in the ovary, Leydig cells in the testis, and sperm cells have high concentrations of circular mitochondrial DNA (mtDNA) [12–14]. In invertebrates, flight muscle tissue and ommatidia in the compound eyes of insects, sperm cells, and gastrodermal cells in the digestive systems of some marine invertebrates, as well as the gills and digestive gland cells in mollusks, are rich in mitochondria and mtDNA [15–17]. Moreover, multiple circular genomes can exist within one mitochondrion.

Similarly, in plants, chlorenchyma cells, especially those found in the mesophyll layer of leaf tissue, contain a significant number of chloroplasts and chloroplast DNA (cpDNA) [18,19].

Increasing evidence is showing that mitochondrial and chloroplast genomes are extensively utilized and highly preferred in various types of research, including genetic diversity and population structure analyses, research on resolving taxonomic ambiguities, divergence-time estimates, haplotype network analyses, molecular metabarcoding, and environmental DNA surveys [20–24]. In addition to conducting their own organelle genome-sequencing studies, researchers have three primary sources for acquiring organelle genomes: the (i) NCBI Organelle Genome Resources (https://www.ncbi.nlm.nih.gov/genome/organelle/ (accessed on 24 October 2023)), (ii) NCBI Nucleotide (https://www.ncbi.nlm.nih.gov/nucleotide/(accessed on 10 November 2023)), and (iii) NCBI Sequence Read Archive (https://www.ncbi.nlm.nih.gov/sra, accessed on 10 November 2023)) databases. Researchers predominantly rely on organellar genome resources and nucleotide databases, often overlooking the massive amount of raw next-generation sequencing data (".fastq") deposited in the NCBI SRA database due to the substantial computational requirements, basic shell scripting, and command-line experience required to manage and process NGS data. Currently, with regard to the experimental aspect, organelle genome-sequencing studies predominantly employ the genome-skimming approach. This approach entails performing low-coverage WGS sequencing and subsequently utilizing bioinformatic tools to extract circular cpDNA and mtDNA genomes from the generated data [25]. This approach is practically favored over the separate isolation of chloroplast and mitochondrial genomes or amplification with long-range PCR. While not as robust as WGS data, RNA-Seq data can also be utilized for the reconstruction of cpDNA and mtDNA genomes [26–28]. This can be attributed to two factors: (i) both rRNA-depleted and poly-A-selected RNA-Seq libraries enable the profiling and discovery of coding and non-coding RNAs, and (ii) the genomic contents of mtDNA and cpDNA primarily consist of coding genes, tRNA genes, and rRNA genes, with only a small fraction dedicated to non-coding intronic and intergenic regions. Several software tools, including GetOrganelle v1.7.7 [29], MitoZ v3.6 [30], and MITGARD v1.0 [26], as well as workflow frameworks like ORTHOSKIM v1.6 [31], go_batch [32], and PhyloHerb v1.1.3 [33], have been developed to facilitate the assembly of organelle genomes and nuclear ribosomal repeats from genomic skimming or RNA-Seq-based transcriptomic data.

Unlike other software and approaches, our workflow stands out due to several distinctive features: (i) We developed a streamlined pipeline for quickly retrieving and organizing SRA data related to multiple species within a particular clade. This process is facilitated using a metafile where users can input the clade's name. Subsequently, data from the SRA for all species belonging to this clade are automatically downloaded and categorized into separate result files corresponding to different data types; (ii) One of our developed scripts, named "*--mt_check* or *--pt_check*", is capable of identifying species lacking mitochondrial or chloroplast genomes but possessing pertinent SRA data through the NCBI database; (iii) The pipeline automatically ranks and streamlines the processing of WGS or RNA-Seq data. It automatically excludes unsuitable NGS data types, such as genotyping-by-sequencing (GBS), RAD-Seq, metabarcoding, and small RNA-Seq data, which are not suitable for obtaining complete mtDNA or cpDNA sequences; (iv) During the automated download process, the pipeline offers users multiple download options, such as sra-tools v3.0.7 (https://github.com/ncbi/sra-tools), IBM Aspera Connect v4.2.6 (https://www.ibm.com/aspera/connect/), and the bash curl (short for "Client URL") command or Efetch v16.0.2 (E-utilities, https://www.ncbi.nlm.nih.gov/home/tools/) to prevent potential download issues; (v) Users can acquire and analyze organellar genomes from SRA WGS and RNA-Seq data without the need to write any code; a single command is sufficient for DNA data, while RNA data require only the execution of two distinct commands.

In this study, we developed an automated bioinformatics workflow designed to unveil high-quality mitochondrial and chloroplast genomes by data mining the NCBI SRA database. Through a case study, we evaluated the effectiveness of our bioinformatics

pipeline. The investigation resulted in the complete revelation of the mitochondrial genome for 76 mollusk species for the first time by mining the SRA database solely by using the keyword "*mollusca*". Utilizing our automated bioinformatics workflow, it becomes feasible to uncover the organelle genomes of numerous species for which genomic or transcriptomic data exist within the SRA database, yet their organelle genomes remain uncharacterized.

## 2. Materials and Methods

### 2.1. Implementation

The OrgaMiner pipeline consists of three fundamental stages: (i) "*download_fastq_from _SRA*", (ii) "*trimming_and_read_quality_assessment*", and (iii) "*assembly_and_annotation_of _organelle_genomes*". In the "*download_fastq_from_SRA*" phase, we obtain the ".fastq" files from the NCBI SRA database using a range of options that will be explained later. Proceeding to the "*trimming_and_read_quality_assessment*" step, its main objectives are to obtain analysis-ready, high-quality NGS ".fastq" reads and to generate summary statistical reports for these reads. We achieve this by applying the trim_galore v0.6.10 (https://www.bioinformatics.babraham.ac.uk/projects/trim_galore/) wrapper for trimming and generating read quality reports for raw NGS reads. All the outputs from trim_galore (with the "*--fastqc*" option) were merged into a single graphic using MultiQC v1.15 [34].

#### 2.1.1. Input File Preparation

To initiate the pipeline, it is necessary to generate a metadata file that contains the scientific names of the target taxa for which organelle genomes are to be extracted. This step facilitates the identification of the relevant SRA files for each taxon for the organelle genome assembly through the utilization of the ESearch utility. Using the "*--mt_check or --pt_check*" option, species for which organelle genomes are already present in the NCBI database can be excluded from the analysis. Alternatively, the metadata file can include both the scientific names of the species and the corresponding accession numbers of the relevant SRA files. For those who want to skip the download process, the metadata file can contain the names of the read files instead of SRA accession numbers. Examples of metadata files can be located in the dedicated GitHub repository at the following link: https://github.com/MolecularBioDiversityLab/OrgaMiner.

#### 2.1.2. Downloading Unprocessed ".fastq" Files from SRA Database

Finding and downloading ".fastq" files of the taxa of interest from the SRA database can be time-consuming for researchers. The OrgaMiner pipeline offers a user-friendly option for selecting the preferred ".fastq" download method when retrieving the relevant SRA files. This feature is facilitated through the use of the Kingfisher-download program (https://wwood.github.io/kingfisher-download/). Subsequently, the SRA accession numbers for paired-end reads are automatically extracted from the metadata file, thereby simplifying the data retrieval process and eliminating the need for users to download the ".fastq" files manually. The primary objective of this step is to acquire organelle genomes from various taxa using the corresponding SRA files available in public databases. However, users also have the option to extract organelle genomes from any WGS or RNA-Seq data that are already stored, as users have the flexibility to skip the download process and initiate the organelle genome assembly process with the "*--skip_download*" option.

#### 2.1.3. Quality Assessment, De Novo Assembly, Annotation, and Outputting

In addition to the GetOrganelle v1.7.7 [29] and MITGARD v1.0 [26] tools, a set of scripts suggested by Senthilkumar et al. [35] was integrated into our pipeline, which enables OrgaMiner to assemble plastid genomes using RNA-Seq data. To ensure that the input reads were suitably prepared for organelle genome assembly using both the GetOrganelle v1.7.7 [29] and MITGARD v1.0 [26] tools, we utilized "*trim_galore*". This Perl wrapper integrates cutadapt v4.4 [36] and FastQC v0.12.1 (https://github.com/s-andrews/FastQC)

and enables adapter trimming and the removal of poly-A tails and undesired reads, and it generates a quality assessment report. To facilitate a more convenient assessment process, the reports were combined and summarized using MultiQC v1.15 [34]. For *de novo* organelle genome assembly, it is essential to ensure that all required arguments are included in the command line, as the pipeline utilizes GetOrganelle v1.7.7 [29] software for genome-skimming data and MITGARD v1.0 [26] software for RNA-Seq data. Users can change the parameters of these software tools in addition to the default settings. While the mitochondrial genomes are annotated using MitoZ v3.6 [30], our pipeline lacks a tool for plastid genome annotation, necessitating the use of external tools for annotating plastid genomes. Upon completing the pipeline, it compiles the ".fastq" files from the SRA database, quality assessment reports, and organelle genomes in the ".fasta" format, and their annotations into distinct directories, each denoted with the appropriate species name. In addition, users can automatically delete the large .fastq files following each assembly to optimize the memory utilization if required. This systematic storage approach ensures convenient access to the output files while minimizing the memory footprint of the pipeline.

### 2.2. Case Studies

The pipeline was devised for the *de novo* assembly of organelle genomes from ".fastq" files from the SRA database encompassing various species that lack representation in the NCBI Organelle Genome Resources or NCBI Nucleotide databases. In this context, the pipeline's configuration file was edited with the addition of the keyword "mollusca". At first, the pipeline automatically collected the scientific names of species that had SRA data from the first step of the pipeline but lacked mitochondrial genomes. By utilizing the Aspera option, we were able to retrieve all appropriate WGS and RNA-Seq data. However, mitochondrial genomes were only obtained for a subset of mollusk species and not for all of them due to issues with the sequence length (shorter than 50 bp) and data quality, quantity, or depth. This was achieved by utilizing the "command_DNA" and "command_RNA" commands with default parameters.

### 2.3. Validation and Application of OrgaMiner Workflow across Diverse Clades

We performed a series of comparative analyses to validate the accuracy of the OrgaMiner workflow's mitochondrial genome annotations. First, we used the workflow to download the NGS data from mollusk species with well-established mitochondrial genome annotations deposited in the NCBI GenBank database. Then, OrgaMiner processed the NGS data and compared the obtained results with those stored in GenBank, demonstrating the reliability of the workflow. The following criteria were used to assess the completeness: the sequence alignment identity (percentage identity), the alignment coverage, and a gene number and composition comparison. Further analyses were conducted to determine the validity of the OrgaMiner workflow in different animal and plant groups. For this reason, the NGS datasets from segmented worms (Phylum Annelida) were selected and analyzed for *de novo* mitochondrial genome assembly and annotation using the OrgaMiner workflow. Similarly, using the same workflow, the NGS datasets belonging to various species within the dicotyledon Solanum and monocotyledon Zingiberales taxonomic groups were assembled and annotated for their chloroplast genomes. Following the assembly and annotation stages using Chlorobox for chloroplast genomes, a comparative analysis was undertaken between the assembled organelle genomes and those obtained from the NCBI. First, a BlastN database was generated with all gene sequences extracted from the reference assemblies. Subsequently, the BlastN ($1 \times 10^{-10}$) algorithm was employed, using the gene sequences of each assembled genome as queries to identify sequence similarities with the reference NCBI genomes. Additionally, the genes present in both the reference genomes and the assembled genomes of each species were compared to assess the completeness of the organelle genomes.

*2.4. Code Availability*

OrgaMiner is implemented in Python and orchestrates a sequence of external tools to extract and identify mitochondrial or chloroplast reads from WGS or RNA-Seq datasets of non-model organisms for which organelle genome sequences are not available but NGS data are deposited in the NCBI SRA database. OrgaMiner is freely available on GitHub (https://github.com/MolecularBioDiversityLab/OrgaMiner) and is compatible with Python version 3.6 or higher. The source code is available under a BSD-3 license. To reproduce all the assemblies generated in this study, please refer to the instructions provided on our GitHub page.

## 3. Results and Discussion

*3.1. Sequence Characteristics of Datasets in Study*

In the context of our research, we developed a flexible workflow that facilitates the automated retrieval, organization, and processing of WGS and RNA-Seq data from the NCBI SRA database to unveil the mitochondrial or chloroplast genome sequences of non-model organisms. We selected the phylum "*mollusca*" to test the practical application of the developed workflow. The phylum Mollusca was also chosen for the following reasons: (i) Mollusca, one of the most diverse animal phyla, accounts for approximately 23% of all known marine species [37]; (ii) these ecologically and economically significant invertebrates inhabit a wide array of marine, freshwater, and terrestrial ecosystems, displaying adaptability to various challenging oceanic environments [38]; (iii) there is a significant amount of NGS data available for the Mollusca phylum in the NCBI SRA database due to the reasons mentioned above. The OrgaMiner pipeline workflow, which consists of two phases, is depicted in Figure 1. First, the user must enter the taxonomic name or both the taxonomic name and the NCBI SRA accession number as input. After obtaining the information, the next step involves searching the NCBI Nucleotide database to download the sequences of species that do not have organelle genomes but do have WGS or RNA-Seq NGS reads in the NCBI SRA database. The second step involves the quality control of the NGS reads, *de novo* organelle genome assembly, and annotation (Figure 1). Upon entering the "*mollusca*" keyword in the workflow config file, the NCBI SRA database was automatically scanned, and the resulting species were matched with the NCBI Nucleotide and Organelle database. The data from species without mtDNA were saved only in ".fastq" format for WGS and RNA-Seq.

According to the mining of the SRA database across the phylum Mollusca, a total of 282 mollusk species possessed NGS data but lacked a corresponding mitochondrial genome. Of these species, 130 have WGS sequencing data, while 152 have RNA-Seq transcriptomic data. Regardless of whether WGS or RNA-Seq data were analyzed, from a taxonomic standpoint, most mollusk species predominantly belong to the Gastropoda, Bivalvia, and Cephalopoda classes (Figure 2). For the *de novo* mitochondrial DNA (mtDNA) analysis of 282 mollusk species, a total of 27.3 billion sequencing reads were obtained, producing over 4 terabases (Tb) of WGS data. Additionally, 9.75 billion sequencing reads were acquired, generating more than 12 Tb of RNA-Seq data, and both datasets were subsequently downloaded and processed (Table S1). While most NGS data were sequenced in the PE150 or PE100 mode with satisfactory Q20/Q30 Phred quality scores, heterogeneity was observed in the NGS data size generated per species. Sequencing data generated in the pair-end sequencing mode constitute more than 97% of the total reads in the WGS dataset and more than 92% in the RNA-Seq dataset (Table S1). The PE reads in the WGS dataset vary from approximately 1 M (million) to 554 M, averaging around 98.6 M reads. In comparison, the PE reads in the RNA-Seq dataset range from about 2.2 M to 308 M reads, averaging approximately 27.9 M reads (Figure 2, Table S1).

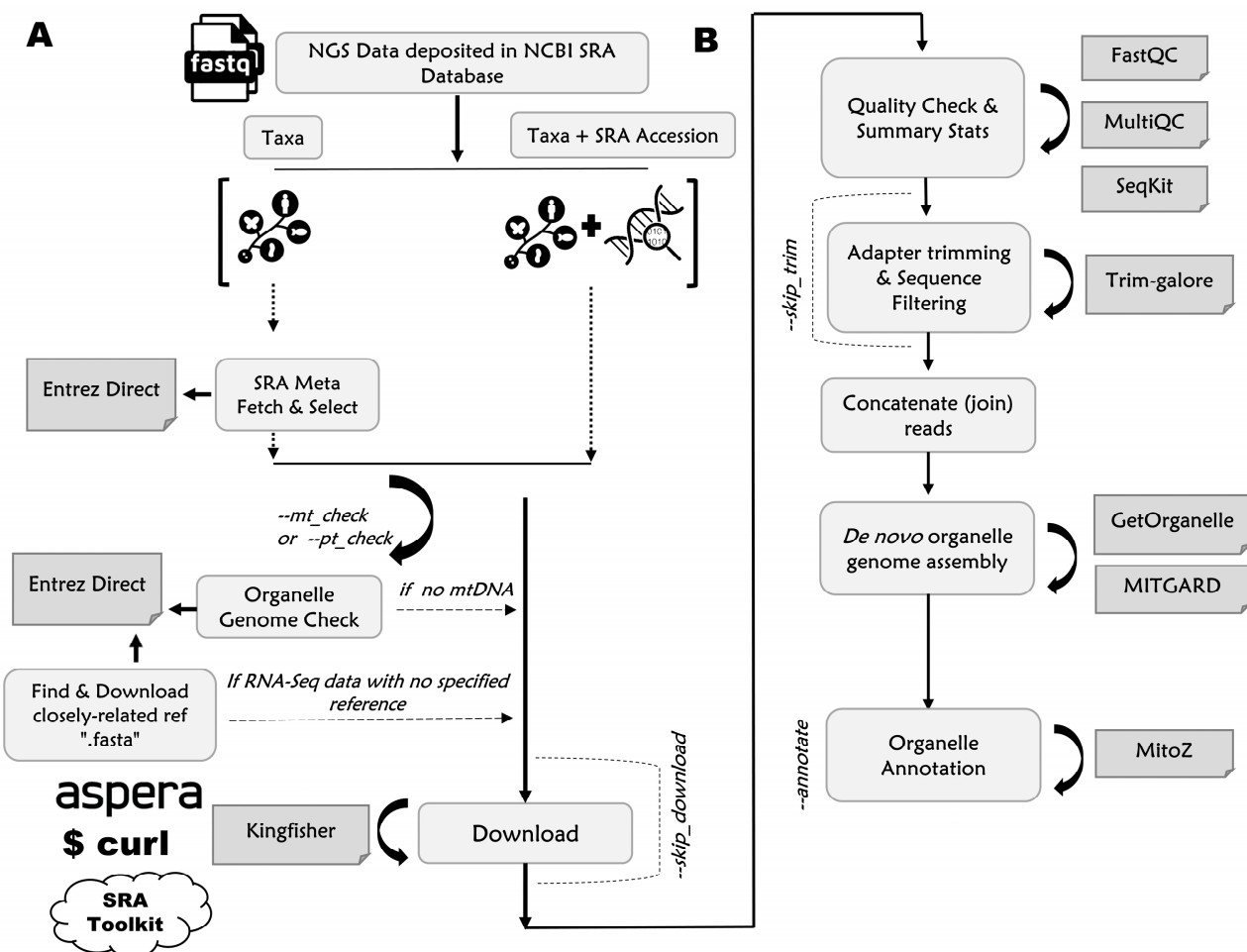

**Figure 1.** OrgaMiner workflow: schematic representation of the NGS data analysis process, from downloading WGS- or RNA-Seq-based ".fastq" files to *de novo* mitochondrial genome assembly and annotations for species with NGS data in the NCBI SRA database but that lack a complete mitochondrial genome. (**A**) The selection of species for *de novo* mtDNA assembly ("--*mt_check* or --*pt_check*") and downloading NGS data via various approaches. (**B**) Subsequent steps involve QC analysis, *de novo* mtDNA assembly, and annotation of the downloaded NGS data.

When examining the NGS sequencing data to be analyzed, it was observed that the read numbers and lengths align with the data sizes reported in the literature. For instance, *Notocrater youngi* (Gastropoda) has around 3.2 million paired-end reads [39]. Another species of Gastropoda, *Planorbella pilsbryi*, has approximately 21 million paired-end reads [40]. Additionally, species such as *Nerita undata* and *Nerita balteata* (Neritimorpha) have produced roughly 5 gigabases (Gb) of paired-end data per sample [41]. The two primary organelle *de novo* genome assembler programs, MitoZ v3.6 [30] (utilizing the assembly module of a modified version of SOAPdenovo-Trans) and GetOrganelle v1.7.7 [29] (employing SPAdes as the assembler), have also suggested that an NGS data yield ranging from 2 to 8 GB is adequate for the organelle genome assembler. Therefore, our WGS- or RNA-Seq-based datasets were considered suitable for the *de novo* assembly and annotation analysis.

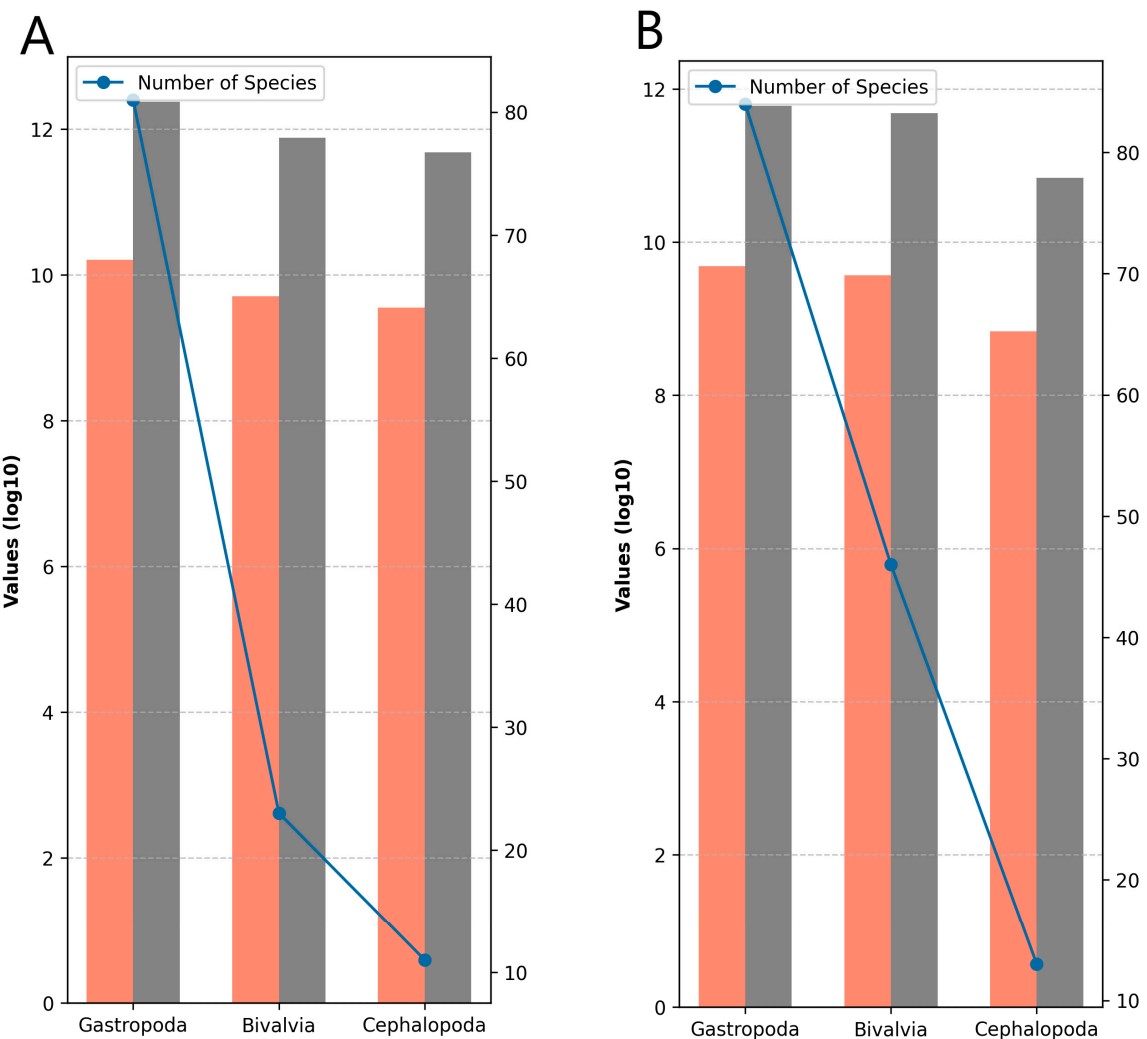

**Figure 2.** Overview of (**A**) WGS dataset and (**B**) RNA-Seq dataset properties and taxonomic distribution of the classes with the most abundant data. Orange bars represent total read numbers, gray bars represent total base numbers, and dots on the blue line represent the numbers of species belonging to these classes. The values on the left side of the graphs represent the logarithm (base 10, $\log_{10}$) of the total read counts and base numbers (bp), relative to the bars. The numbers on the right side of the graphs are related to the lines and represent the numbers of species.

### 3.2. Molluscan mtDNA Annotation Findings and Implications

Although species belonging to the phylum Mollusca generally exhibit structural compatibility with ancestral mitochondria in terms of the mitochondrial DNA organization (typically encoding 13 proteins, two rRNAs, and 22 tRNAs, as well as a putative control region), they display extraordinary variation in size and architecture within the animal kingdom, such as radical genome rearrangements, gene duplications and losses, and the introduction of novel genes [42–45]. Despite their complex mitochondrial structure, a high-quality mtDNA sequence and annotation were obtained by mining the NCBI database for 76 mollusk species for which an mtDNA sequence has not yet been revealed. Of these complete mtDNAs, 52 were acquired through WGS analysis (Table 1), while 24 were obtained through RNA-Seq read processing (Table 2). Among bilateral animals, the mitochondrial genomes of the phylum Mollusca exhibit considerable size variation [46]. Upon an examination of 3396 mitochondrial genomes in the NCBI Nucleotide database, it was observed that the minimum length of the mtDNA genome was 13.1 kb (*Pliocardia stearnsii*), while the maximum was 67.1 kb (zebra mussel, *Dreissena polymorpha*), with an average length of 16.2 kb. In Tables 1 and 2, we find that the average length of the mtDNA genomes was

approximately 16 kb. The average length of the mtDNA genomes obtained through *de novo* read assembly in the WGS and RNA-Seq datasets was consistent with the literature and mitochondrial genome database for Mollusca [37].

**Table 1.** Basic information on the mitochondrial genome characteristics of various mollusk species, including the mtDNA genome lengths and numbers of coding and non-coding genes, obtained through the WGS data in the NCBI SRA database. * indicates that the ND3 gene was identified both in the H-strand and L-strand during the annotation processes.

| Class | Family | Species | Coding Genes | tRNA Genes | rRNA Genes | Missing Genes | Total Genes | mtDNA Length |
|---|---|---|---|---|---|---|---|---|
| Bivalvia | Mytilidae | *Botula fusca* | 12 | 21 | 2 | 2 | 35 | 19,595 |
| Bivalvia | Unionidae | *Elliptio hopetonensis* | 13 | 22 | 2 | 0 | 37 | 15,775 |
| Bivalvia | Tellinidae | *Macoma nasuta* | 12 | 22 | 2 | 1 | 36 | 17,348 |
| Bivalvia | Unionidae | *Megalonaias nervosa* | 13 | 22 | 2 | 0 | 37 | 16,026 |
| Bivalvia | Anomiidae | *Pododesmus macrochisma* | 13 | 22 | 2 | 0 | 37 | 15,080 |
| Bivalvia | Veneridae | *Saxidomus gigantea* | 13 | 22 | 2 | 0 | 37 | 19,754 |
| Cephalopoda | Octopodidae | *Muusoctopus eicomar* | 13 | 22 | 2 | 0 | 37 | 16,168 |
| Cephalopoda | Octopodidae | *Muusoctopus leioderma* * | 13 | 22 | 2 | 0 | 37 | 17,006 |
| Cephalopoda | Octopodidae | *Muusoctopus longibrachus* | 13 | 22 | 2 | 0 | 37 | 16,192 |
| Cephalopoda | Octopodidae | *Octopus americanus* | 13 | 22 | 2 | 0 | 37 | 15,655 |
| Cephalopoda | Octopodidae | *Amphioctopus burryi* | 13 | 22 | 2 | 0 | 37 | 15,883 |
| Cephalopoda | Sepiolidae | *Rondeletiola minor* | 13 | 22 | 2 | 0 | 37 | 15,800 |
| Cephalopoda | Loliginidae | *Doryteuthis pealeii* | 13 | 21 | 2 | 1 | 36 | 16,674 |
| Gastropoda | Onchidorididae | *Corambe burchi* | 13 | 22 | 2 | 0 | 37 | 14,308 |
| Gastropoda | Neomphalidae | *Cyathermia naticoides* * | 13 | 22 | 2 | 0 | 37 | 16,156 |
| Gastropoda | Ovulidae | *Cyphoma gibbosum* | 13 | 20 | 2 | 2 | 35 | 16,638 |
| Gastropoda | Dironidae | *Dirona albolineata* | 13 | 23 | 2 | 0 | 38 | 14,651 |
| Gastropoda | Dorididae | *Doris verrucosa* * | 13 | 22 | 2 | 0 | 37 | 14,518 |
| Gastropoda | Plakobranchidae | *Elysia diomedea* | 13 | 21 | 2 | 1 | 36 | 14,158 |
| Gastropoda | Chromodorididae | *Goniobranchus kuniei* | 13 | 23 | 2 | 0 | 38 | 14,738 |
| Gastropoda | Haliotidae | *Haliotis corrugata* | 13 | 22 | 2 | 0 | 37 | 16,951 |
| Gastropoda | Haliotidae | *Haliotis discus discus* | 13 | 22 | 2 | 0 | 37 | 16,805 |
| Gastropoda | Haliotidae | *Haliotis fulgens* | 13 | 22 | 2 | 0 | 37 | 16,376 |
| Gastropoda | Haliotidae | *Haliotis gigantea* | 13 | 22 | 2 | 0 | 37 | 16,539 |
| Gastropoda | Haliotidae | *Haliotis kamtschatkana* | 13 | 22 | 2 | 0 | 37 | 16,892 |
| Gastropoda | Haliotidae | *Haliotis madaka* | 13 | 22 | 2 | 0 | 37 | 16,745 |
| Gastropoda | Haliotidae | *Haliotis midae* | 13 | 22 | 2 | 0 | 37 | 16,530 |
| Gastropoda | Haliotidae | *Haliotis sorenseni* | 13 | 22 | 2 | 0 | 37 | 16,711 |
| Gastropoda | Glaucidae | *Hermissenda crassicornis* | 13 | 21 | 2 | 1 | 36 | 14,750 |
| Gastropoda | Lepetodrilidae | *Lepetodrilus galriftensis* * | 13 | 22 | 2 | 0 | 37 | 19,339 |
| Gastropoda | Lepetodrilidae | *Lepetodrilus gordensis* | 13 | 22 | 2 | 0 | 37 | 16,455 |
| Gastropoda | Littorinidae | *Littorina arcana* | 13 | 22 | 2 | 0 | 37 | 16,301 |
| Gastropoda | Littorinidae | *Littorina compressa* | 13 | 22 | 2 | 0 | 37 | 16,349 |
| Gastropoda | Lottiidae | *Lottia persona* | 12 | 22 | 2 | 1 | 36 | 17,106 |
| Gastropoda | Peltospiridae | *Peltospira delicata* | 13 | 23 | 2 | 0 | 38 | 15,523 |
| Gastropoda | Tateidae | *Potamopyrgus kaitunuparaoa* | 13 | 23 | 2 | 0 | 38 | 15,332 |
| Gastropoda | Lepetodrilidae | *Pseudorimula midatlantica* * | 13 | 22 | 2 | 0 | 37 | 16,411 |
| Gastropoda | Lymnaeidae | *Radix swinhoei* | 13 | 19 | 2 | 3 | 34 | 14,998 |
| Gastropoda | Scyllaeidae | *Scyllaea pelagica* | 13 | 22 | 2 | 0 | 37 | 14,759 |
| Gastropoda | Lymnaeidae | *Stagnicola palustris* | 12 | 21 | 2 | 2 | 35 | 14,261 |
| Gastropoda | Tritoniidae | *Tritonia tetraquetra* | 13 | 22 | 2 | 0 | 37 | 15,087 |
| Gastropoda | Tylodinidae | *Tylodina fungina* | 13 | 21 | 2 | 2 | 36 | 14,649 |

**Table 1.** *Cont.*

| Class | Family | Species | Coding Genes | tRNA Genes | rRNA Genes | Missing Genes | Total Genes | mtDNA Length |
|---|---|---|---|---|---|---|---|---|
| Gastropoda | Aegiretidae | *Aegires albopunctatus* | 13 | 21 | 3 | 1 | 37 | 13,947 |
| Gastropoda | Aeolidiidae | *Aeolidia papillosa* | 12 | 22 | 2 | 2 | 36 | 16,696 |
| Gastropoda | Goniodorididae | *Ancula gibbosa* | 13 | 22 | 2 | 0 | 37 | 14,532 |
| Polyplacophora | Mopaliidae | *Mopalia ciliata* * | 13 | 21 | 2 | 1 | 36 | 13,987 |
| Polyplacophora | Mopaliidae | *Mopalia kennerleyi* * | 13 | 22 | 2 | 0 | 37 | 14,290 |
| Polyplacophora | Mopaliidae | *Mopalia muscosa* * | 13 | 22 | 2 | 0 | 37 | 14,976 |
| Polyplacophora | Mopaliidae | *Mopalia swanii* * | 13 | 22 | 2 | 0 | 37 | 14,969 |
| Polyplacophora | Mopaliidae | *Mopalia vespertina* * | 13 | 22 | 2 | 0 | 37 | 14,987 |
| Polyplacophora | Chitonidae | *Acanthopleura granulata* | 13 | 22 | 2 | 0 | 37 | 15,618 |
| Solenogastres | Gymnomeniidae | *Wirenia argentea* | 13 | 20 | 2 | 2 | 35 | 16,443 |

**Table 2.** Overview of mtDNA genome lengths and compositions of some mollusk species analyzed through RNA-Seq data in the NCBI SRA database. [†] indicates that the ND3 gene was identified both in the H-strand and L-strand during the annotation processes.

| Class | Family | Species | Coding Genes | tRNA Genes | rRNA Genes | Missing Genes | Total Genes | mtDNA Length |
|---|---|---|---|---|---|---|---|---|
| Bivalvia | Mactridae | *Mactra antiquata* | 13 | 18 | 2 | 5 | 33 | 16,429 |
| Bivalvia | Unionidae | *Uniomerus tetralasmus* | 13 | 18 | 2 | 4 | 33 | 15,247 |
| Bivalvia | Mytilidae | *Mytilus planulatus* | 13 | 22 | 2 | 1 | 37 | 16,727 |
| Bivalvia | Vesicomyidae | *Archivesica packardana* [†] | 14 | 22 | 2 | 0 | 38 | 16,467 |
| Bivalvia | Pharidae | *Ensis directus* | 13 | 23 | 2 | 0 | 38 | 16,925 |
| Bivalvia | Ostreidae | *Saccostrea palmula* | 13 | 19 | 2 | 4 | 34 | 16,130 |
| Bivalvia | Mytilidae | *Gigantidas horikoshii* | 12 | 20 | 2 | 3 | 34 | 17,504 |
| Bivalvia | Thyasiridae | *Conchocele bisecta* | 12 | 22 | 2 | 2 | 36 | 17,181 |
| Cephalopoda | Sepiolidae | *Rossia pacifica* [†] | 14 | 18 | 2 | 4 | 34 | 14,897 |
| Cephalopoda | Octopodidae | *Enteroctopus megalocyathus* [†] | 14 | 20 | 2 | 2 | 36 | 16,027 |
| Gastropoda | Ranellidae | *Monoplex corrugatus* | 13 | 18 | 2 | 4 | 33 | 16,178 |
| Gastropoda | Planorbidae | *Biomphalaria alexandrina* | 13 | 19 | 2 | 3 | 34 | 13,570 |
| Gastropoda | Conidae | *Conus ammiralis* | 13 | 19 | 2 | 3 | 34 | 15,459 |
| Gastropoda | Conidae | *Conus purpurascens* | 13 | 19 | 2 | 3 | 34 | 15,509 |
| Gastropoda | Facelinidae | *Facelina rubrovittata* | 13 | 19 | 2 | 3 | 34 | 14,481 |
| Gastropoda | Chromodorididae | *Verconia verconis* | 13 | 19 | 2 | 3 | 34 | 14,560 |
| Gastropoda | Nacellidae | *Cellana rota* [†] | 14 | 19 | 2 | 3 | 35 | 16,042 |
| Gastropoda | Semisulcospiridae | *Semisulcospira reiniana* | 13 | 20 | 2 | 4 | 35 | 15,291 |
| Gastropoda | Tritoniidae | *Tritoniopsis frydis* [†] | 13 | 20 | 2 | 5 | 35 | 14,481 |
| Gastropoda | Turbinidae | *Angaria nodosa* | 14 | 20 | 2 | 3 | 36 | 19,389 |
| Gastropoda | Conidae | *Conus bayani* [†] | 14 | 20 | 2 | 2 | 36 | 15,525 |
| Gastropoda | Nacellidae | *Nacella polaris* | 13 | 21 | 2 | 1 | 36 | 16,752 |
| Gastropoda | Conidae | *Conus chaldaeus* | 13 | 22 | 2 | 0 | 37 | 15,442 |
| Polyplacophora | Chitonidae | *Tonicia schrammi* | 14 | 18 | 2 | 4 | 34 | 14,909 |

In the WGS dataset, the Gastropoda class had the highest number of mitochondrial DNA (mtDNA) genomes (32), followed by Cephalopoda (7), Bivalvia (6), and Polyplacophora (6). Among the RNA-Seq dataset, Gastropoda had the highest number of mitochondrial DNA (mtDNA) genomes with 13, followed by Bivalvia with 8, Cephalopoda with 2, and Polyplacophora with 1. The mtDNA genomes constructed from the WGS dataset show a significantly low count of missing genes, and the counts of coding and non-coding genes are consistent with the general mitochondrial genome pattern observed in mollusks. The presence of 16S large subunit rRNA and 12S small subunit rRNA was thoroughly identified

in the *de novo* assembled mtDNA genomes, regardless of the taxonomic classification. In some bivalve species, the absence of the *Atp8* gene leads to a reduction in the number of coding genes to 12 [46–48], while, in other mollusk species, there are 13 coding genes, in line with the ancestral mollusk genome. When examining Table 1, it becomes apparent that three Gastropoda species (*Lottia persona*, *Stagnicola palustris*, and *Aeolidia papillosa*) possess 12 coding genes, and upon characterizing the missing gene, it was identified as the *Atp8* gene. Later, it was elucidated that this circumstance is not attributed to the absence of the *Atp8* gene in the mtDNA genome, observed in the class Bivalvia, but rather to challenges in annotating this particular gene. In recent studies, the annotation of the *Atp8* gene has posed challenges due to its high variation and short length [42,49]. We also need to emphasize the following regarding protein-coding genes: in Tables 1 and 2, some species are marked with * and †, and these marks indicate that the NADH dehydrogenase subunit 3 (*Nad3*) gene is annotated on both the heavy (high G + T content) and light (low G + T content) strands in these species. We believe that this issue arises from challenges in annotating molluscan genomes. Because of the transcription of mtDNA as polycistronic RNA, it is considered physically impossible to have gene overlap between two protein-coding genes encoded on the same strand and in the same open reading frame, but it is possible if the frames are different [42]. Secondly, the boundaries of some coding genes (correct start and stop codon locations) cannot be determined precisely with the current annotation tools following the *de novo* assembly of NGS short reads, and this may lead to incorrect annotations. Therefore, third-generation technologies, such as Oxford Nanopore or PacBio, may be more effective in mtDNA genome assembly and annotation [50,51]. The mtDNA genome of a typical mollusk contains 22 transfer RNA (tRNA) genes. The mean number of recovered tRNA genes in the mtDNA genomes generated from the WGS analysis was 21.8 (Table 1). However, some tRNA genes could not be annotated in the mtDNA genome produced from the RNA-Seq dataset, resulting in an average number of 19.8 tRNA genes (as shown in Table 2). Recent studies suggest that changes and duplications in mitochondrial tRNA genes contribute significantly to the rearrangement of the mitochondrial genome [42,52]. Our analysis showed that some mollusk species might exhibit duplications in tRNA genes, resulting in over 22 tRNA genes, while fewer than 20 mitochondrial tRNA genes generally indicate an annotation issue. All GenBank annotations (".gb" and ".gbk" files) and circular mitochondrial DNA (mtDNA) plots generated through the analysis of WGS and RNA-Seq data are included in the Supplementary File. To evaluate the accuracy of the mollusk mitochondrial genome annotations generated by the OrgaMiner workflow, NGS data from species with known mitochondrial genome sequences and annotations were analyzed using the workflow. The resulting mitochondrial genome and annotation files were then compared to the corresponding data available in the NCBI Nucleotide database. These analyses demonstrated that the OrgaMiner workflow produces mitochondrial genomes that are highly similar to those in the NCBI Nucleotide database, as evidenced by metrics such as the alignment identity (percentage identity), the alignment coverage, and gene number/composition comparisons. We processed totals of 10 pieces of WGS and 10 pieces of RNA-Seq mollusk data using the OrgaMiner workflow and compared the results to their mitochondrial genomes in the NCBI Nucleotide database (Table S3). Compared to the reference mtDNA, regardless of coding or non-coding genes, the alignment identities were found to be >99.18% for the WGS data and >98.60% for the RNA-Seq data. However, the WGS data appear to be more successful at extracting mitochondrial genes than the RNA-Seq data ("genome comparison" in Table S3). The validation results largely indicate the reliability of the OrgaMiner workflow in analyzing NGS data from the NCBI SRA. In some cases, partial mitochondrial and chloroplast genomes may be obtained from the OrgaMiner workflow. This occurrence could be attributed to the nature of the RNA-Seq or WGS NGS datasets in the NCBI SRA database rather than to the OrgaMiner workflow itself. While the read lengths, depths, and Phred quality scores are satisfactory, several factors inherent to NGS data in the ".fastq" format could negatively impact the analysis. These factors include the following: (i) the low levels of organellar coding genes, and particularly

tRNA genes, represented in RNA-Seq data, unlike in WGS data; (ii) the high duplication rates of certain NGS reads, such as the high rRNA duplication common in RNA-Seq data; and (iii) the potential presence of contaminant sequences (non-target organisms).

Despite the typically high copy numbers of mitochondria (and, hence, mtDNAs) in tissues, fewer organelle genomes and coding/non-coding genes were detected in the RNA-Seq data compared to the WGS dataset. This discrepancy could be attributed to various potential reasons. First, mitochondrial transcripts are polyadenylated by MTPAP (polyadenylic acid RNA polymerase), adding 40–50 adenine nucleotides to almost all mitochondrial transcripts, which is crucial for mRNA stability and regulation [53,54]. The poly-A tail lengths of nuclear mRNA transcripts vary depending on the tissue type, and the median tail lengths of nuclear mRNA transcripts are generally longer (~250 nucleotides) than those of mitochondrial transcripts [55,56]. Nuclear mRNA transcripts might also be more easily captured in RNA-Seq libraries, which are prepared from total RNA through the poly(A) enrichment of mRNA (mRNA-Seq). In addition to coding genes, one notable category of genes often overlooked and challenging to predict in RNA-Seq analysis comprises tRNA genes. Due to their shorter length and lack of polyadenylation, tRNA genes are less frequently represented and sequenced than mRNA and rRNA genes in both poly-A-captured and rRNA-depleted libraries. Moreover, during the bioinformatics preprocessing step, sequences with fewer than 50 bases in the PE100 and PE150 .fastq files are typically filtered out and excluded from the analysis. tRNA genes are easily captured in small RNA-Seq libraries, but such libraries cannot be used in *de novo* mitochondrial genome assembly, so they are excluded from our OrgaMiner workflow. In spite of our numerous attempts to cover all genes, we obtained a total of 132 mtDNA partial genomes from the WGS (50) and RNA-Seq (82) datasets of various mollusk species, even though their mtDNA quality appeared to be low. Despite this, these mtDNA sequences can be helpful in population genetics, haplotype analysis, and molecular barcoding studies. Therefore, their sequence and annotations are included in Table S2.

To test whether complex mitogenomic rearrangements were correctly annotated using OrgaMiner, special attention was given to Pectinidae, one of the most peculiar groups within Mollusca. While gene rearrangements are most common for tRNAs among metazoans in general, the mitochondrial genome rearrangements in Pectinidae often involve coding genes, unlike what has been observed in other animal groups. Malkócs et al. [57] investigated the mitogenomic rearrangements observed in certain Pectinidae species and visualized their mitochondrial structural variations. Using the OrgaMiner workflow, we obtained annotations for three species within this family and compared them with published data (Figure 6 in Malkócs et al. [45]). The order and orientation of the gene arrangement pattern are identical to that of the study. This comparison and the mitochondrial rearrangements shown in Figure 3 validate the ability of the workflow to annotate mitochondrial structural variations as well.

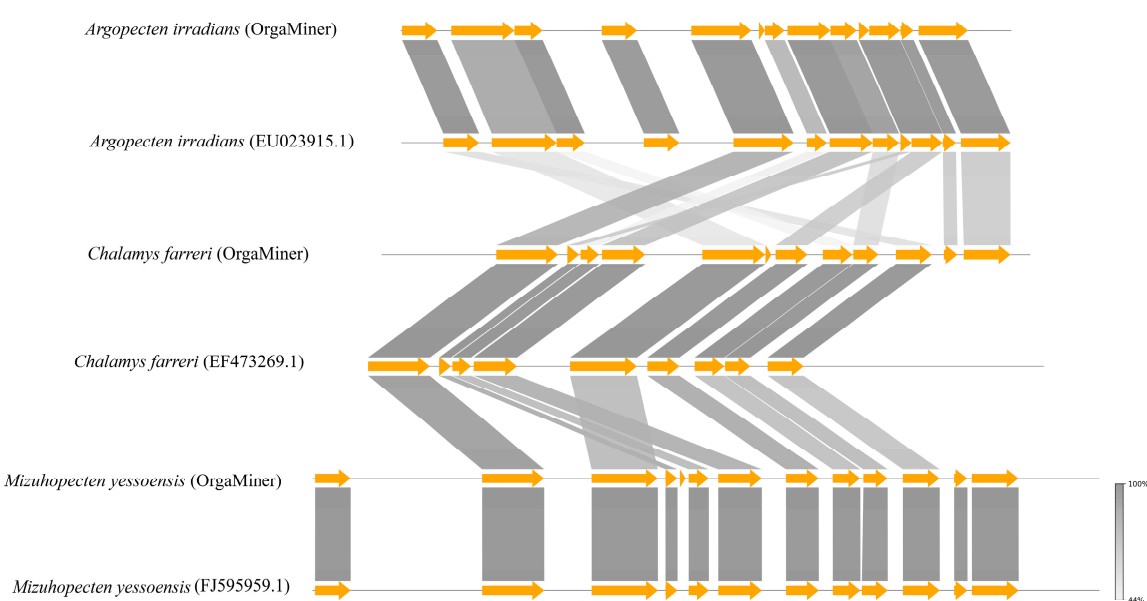

**Figure 3.** Structural and synteny comparisons of mitochondrial gene rearrangements observed in Pectinidae species were performed by uploading annotations of mtDNAs using pyGenomeViz (https://github.com/moshi4/pyGenomeViz) with default settings.

### 3.3. Applicability of OrgaMiner Workflow across Different Clades

To test the validity of the OrgaMiner workflow for other organism groups, we performed mitochondrial genome analyses on animals with the taxonomic keyword Annelida. As for plants, we analyzed the chloroplast genomes of both dicotyledons from the Solanum genus and monocotyledons from the Zingiberales order. A total of 37 annelid mitochondrial genomes and annotations were obtained, 27 from whole-genome-sequencing (WGS) data and 10 from RNA-Seq data. When examining the mitochondrial genome assembly and annotations of these species in the NCBI Nucleotide database, it can be seen that accurate results were obtained using the OrgaMiner workflow (Table S4). Compared to the reference mitogenomes, for annelid species, the alignment identities were >99.65% for the WGS data and >96.67% for the RNA-Seq data. Using the OrgaMiner workflow, nearly all the mitochondrial genes of annelids were comprehensively covered. As observed in the mollusk species, the WGS data appeared to outperform the RNA-Seq data in the de novo retrieval and annotation of annelid organelle genomes regarding the number of genes retrieved ("mitochondrial comparison" in Table S4). As for plant species, we analyzed 15 species from the Solanum genus, including 6 with WGS data and 9 with RNA-Seq data, as well as 26 species from the Zingiberales order, all with WGS data. The results from both groups were compared to their reference chloroplast genome sequence and annotations. Regardless of the plant species, all species showed over 96% alignment identity. In the WGS data analysis of species from the genus Solanum and the order Zingiberales, the chloroplast genome and genes are largely represented and compatible with the reference genome/annotation ("plastid comparison" in Table S4). Although the de novo-assembled chloroplast genomes from both the WGS and RNA-Seq data showed promising results, with many unique genes aligned and commonly found in the reference genomes, there were some inconsistencies observed between the reference genomes and OrgaMiner-assembled genomes, as the gene numbers and compositions differed more than expected. This difficulty may have arisen due to the challenging nature of annotating plastid genomes, a task that requires manual curation, which users may contemplate following the assembly of plastid genomes [57].

### 3.4. Liminations of OrgaMiner Pipeline and Recommendations

OrgaMiner is a tool that quickly searches for species without organelle genomes (mtDNA or cpDNA) in the NCBI Nucleotide database, and it obtains organelle genomes by processing WGS and RNA-Seq data from the NCBI SRA database for species for which

organelle genomes are not yet known. Although useful for quickly revealing new organelle genomes, there are factors limiting the effectiveness of this workflow for mining the database. A current limitation of these tools is that the pipeline's data acquisition process often necessitates substantial storage resources, particularly when dealing with taxa encompassing large numbers of species. To ameliorate this weakness slightly, users can use the "*--remove*" option for removing ".fastq" files or the "*--remove-all*" option, which deletes all ".fastq" files, including those downloaded or already stored, following each assembly process. Nevertheless, the storage requirement remains a significant concern, potentially posing challenges for users with limited storage capacities. Secondly, when focusing on plastid genome analysis, the OrgaMiner exhibits inherent limitations. One notable constraint is its inability to derive plastid genomes from RNA-Seq data, requiring users to resort to alternative methods for plastid genome reconstruction from transcriptome data. Additionally, the absence of plastid genome annotation functionality within the pipeline necessitates the use of external annotation tools and databases, introducing potential additional steps. Furthermore, the efficiency of the pipeline is influenced by the choice of download options. While alternative download methods may offer relative speed advantages compared to the default sra-tools, the pipeline may encounter challenges in acquiring FTP links, potentially impeding data retrieval in some instances.

## 4. Conclusions

Taken together, through extensive analysis and evaluations on real ".fastq" data, we report a user-friendly bioinformatics pipeline called OrgaMiner, which enables the management, exploitation, and mining of large genomic and transcriptomic datasets available in the NCBI SRA database to uncover high-quality mitochondrial and chloroplast genomes for non-model organisms automatically. By successfully demonstrating its utility in the *de novo* assembly of mitochondrial genomes for various mollusk species solely through the keyword "mollusca," we provide a valuable resource for researchers working with taxonomic and population genetics questions. The versatility of the pipeline extends its applicability to diverse species, including invertebrates, vertebrates, and plants, making it an accessible and efficient solution for organelle genome assembly used mainly in molecular taxonomy, population genetics, and haplotype network analysis.

**Supplementary Materials:** The following supporting information can be downloaded at https://www.mdpi.com/article/10.3390/d16020104/s1, Table S1. The sequencing summary stats and basic SRA metainformation for the ".fastq" files used in this study; Table S2. The basic annotations and sequences of various mollusk species whose mitochondrial genomes were not entirely covered in the WGS or RNA-Seq datasets; Table S3. Comparative analysis of mollusk mitochondrial genome sequences generated with OrgaMiner workflow against their reference mitochondrial genomes from NCBI Nucleotide database; Table S4. The basic organelle genome features determined by OrgaMiner workflow in animal species belonging to Annelida and plant species belonging to the *Solanum* genus and Zingiberales order.

**Author Contributions:** Conceptualization, V.E.; methodology, V.E. and M.A.B.; pipeline development and implementation, M.A.B. and V.E.; validation, V.E. and M.A.B.; writing—review and editing, V.E.; visualization, M.A.B. and V.E.; funding acquisition, V.E. All authors have read and agreed to the published version of the manuscript.

**Funding:** This study was funded by the Scientific Research Projects Coordination Unit of Istanbul University, grant number 37223. The computing resources used in this work were funded by the National Center for High Performance Computing of Turkey (UHeM) under grant number 5004732017.

**Institutional Review Board Statement:** Not applicable.

**Data Availability Statement:** All relevant data presented in the study are included in the article and Supplementary Material. The source codes (Python and BASH scripts) along with detailed instructions on how these must be executed are also freely available in the GitHub repository (https://github.com/MolecularBioDiversityLab/OrgaMiner).

**Acknowledgments:** We express our gratitude to Tuana Öğretici for her invaluable assistance with developing the Python scripts and the subsequent transfer of the fully functional code to the GitHub page of our Molecular Biodiversity Lab. We also thank the members of our laboratories for the helpful comments. We are also grateful to the anonymous reviewers for their valuable comments on the manuscript.

**Conflicts of Interest:** The authors declare no conflicts of interest. The funders had no role in the design of the study; in the collection, analyses, or interpretation of the data; in the writing of the manuscript; or in the decision to publish the results.

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
