# Peer review of "Mining NCBI Sequence Read Archive Database: An Untapped Source of Organelle Genomes for Taxonomic and Comparative Genomics Research"

_diversity, doi:10.3390/d16020104_

Round 1
Reviewer 1 Report
Comments and Suggestions for Authors
The present study developed an automated bioinformatics workflow to unveil high-quality mitochondrial and chloroplast genomes through data mining the NCBI SRA database. The present study could provide an easy way for researchers working on taxonomy and molecular phylogenies. In my opinion, this manuscript could be accepted for publication. I also have some concerns.
1. I have checked OrgaMiner on the website. It seems that SPAdes was the major software for mitochondrial and chloroplast genome assembly from genomic data. I was wondering why the authors did not try other softwares? According to my own experience, different softwares could probably generated different results.
2. How did the authors evaluate the quality of the assembled mitogenomes?
3. Line 178: One "sequence" should be deleted.
4. Line 293: Here "two species" might be changed to "three species" since three names were listed in the brackets.
Reviewer 2 Report
Comments and Suggestions for Authors
In the manuscript "Mining NCBI Sequence Read Archive database: An untapped source of organelle genomes for taxonomic and comparative genomics research", the authors present OrgaMiner, a new tool supposed to facilitate the mining of organelle genomes from the NCBI SRA database. Unlike existing tools, OrgaMiner is focused on user-friendliness; its usage does not require virtually any coding skills. It can also handle vast quantities of data, automatically searching for results, downloading them, assessing their quality and processing them into annotated genomes. At first glance, the tool looks like an impressive and substantial advancement; however, at second glance, the study has significant caveats that have to be fixed before it can be published.
First and most importantly, OrgaMiner testing lacks any control. Whereas authors assembled and annotated an impressive number of mitochondrial genomes, it is unclear how far these assemblies and annotations are correct. Authors should test its accuracy using one of two methods:
1) assemble and annotate a set of mitochondrial genomes in OrgaMiner, then assemble and annotate the same set manually using standard methods; present the results next to each other to show discrepancies between both annotations
2) use OrgaMiner to assemble and annotate genomes that were already published, then present results from OrgaMiner next to the previously published results
Second, the authors claim that OrgaMiner works universally for any organelles, but they have tested it only with mitochondrial genomes of one group of organisms. I suggest that at least two additional tests should be performed to prove that it works as universally as the manuscript states:
1) It needs to be tested with genomes of plastids. The manuscript gives the impression that the authors tried it; it is just not presented - if this is the case, please add it to the manuscript.
2) Some organisms have mitochondrial genomes that are more variable than molluscs (for instance, see the paper attached). Mitochondrial chromosomes can be fragmented and linear; cases of heteroplasmy are also known. Authors acknowledge that mitochondria may contain multiple chromosomes (lines 44-45), but they do not always have to be circular, and it is not clear how well OrgaMiner works in these cases. Please test it in some animals with fragmented mitogenomes or linear mitochondrial chromosomes, or if it is not targeted for these groups, clearly state it in the manuscript.
It goes without saying that the tests with plastids and fragmented mitogenomes should be performed with control, as explained in the first point.
Third, according to my experience, MitoZ is notoriously inaccurate in annotating protein-coding genes; I have experienced many cases similar to what the authors describe with the Atp8 gene. A possible solution is to verify the annotation by searching open reading frames; I suppose it might not be that hard to build this step into OrgaMiner. Did the authors consider this? The manuscript should discuss this possibility since it might significantly improve annotation quality (it also eliminates many overlaps).
Besides these central issues, I also have some minor suggestions, see below:
Lines 147-149 - Is it that hard to build NOVOPlast directly into OrgaMiner so that it also works for the plastids? I have no experience with NOVOPlast, so it is just a suggestion.
Line 178 - Has the length of 50 bp proved suitable? Does it perform somehow differently than, e.g. in 80 bp? It might be helpful if OrgaMiner had the option to set this length manually.
Lines 198-200 - I think the diverse lifestyles are not very relevant; it is more important how diverse their mitogenomes are. OrgaMiner should be tested in more extreme variations of mitogenomes than molluscs; see above. However, molluscs exhibit some unusual mitochondrial gene rearrangements, also mentioned in lines 256-261. How well did OrgaMiner perform in annotating these duplicated and rearranged genes? This is of considerable interest and should be commented on more clearly and in detail in the manuscript.
Line 226 – Gastropoda, Bivalvia, and Cephalopoda are not families but classes. Please correct this.
Lines 234-237 - Excessive coverage of mitochondrial genomes can lead to assembly errors (own experience). OrgaMiner enables subsampling (--max-reads option), which is not commented on in the manuscript. Did the authors test it, and what were the results? This might be substantially important for the results and should be at least mentioned in the manuscript.
Line 265 - should be "bilateral" instead of "bilaterian"
Line 267 - should be "mollusc" instead of "Mollusk"
Tables 1+2 - Consider adding a column with a family between "Class" and "Species". Mitogenomes of molluscs are diverse, and having the family on hand would facilitate comparison with literature.
Tables 1+2 - What the column "Missing genes" means is unclear. Are these genes absent in the particular species (i.e., they are published in the literature that the species does not have them) or genes that should be present but OrgaMiner did not find them? The same for the other columns; it is not clear how many genes should be present according to the literature and how many of these OrgaMiner managed to annotate. This is important because it shows how accurate the annotation is. Please make this clear throughout the tables (also in the supplement).
Line 341 - What does the "X" mean? Please correct.
Lines 352-357 - Storage Requirements - How does OrgaMiner download the data? Does it download all of them at once and analyze them, or does it download one sample, analyze it, delete it, and then download the next? I am asking because the latter way is substantially less storage-demanding. This is not clearly explained in the manuscript; please clarify.
Unpublished data (mitogenome gbf files) - these files are obviously from the MitoZ paper of Meng et al. (2019). It would also be helpful to have the annotations prepared by OrgaMiner to compare them and see how they match.

Round 2
Reviewer 2 Report
Comments and Suggestions for Authors
My main objection was that the results of OrgaMiner were not tested against control. Since the authors tested it, I think it looks substantially better, and it might be published, provided the editor does not have doubts that everything the authors claim is true.